# The Effect of Chair-Based Exercise on Physical Function in Older Adults: A Systematic Review and Meta-Analysis

**DOI:** 10.3390/ijerph18041902

**Published:** 2021-02-16

**Authors:** Natalie Klempel, Nicole E. Blackburn, Ilona L. McMullan, Jason J. Wilson, Lee Smith, Conor Cunningham, Roger O’Sullivan, Paolo Caserotti, Mark A. Tully

**Affiliations:** 1Institute of Nursing and Health Research, School of Health Sciences, Ulster University, Newtownabbey BT37 0QB, UK; Klempel-N@ulster.ac.uk (N.K.); ne.blackburn@ulster.ac.uk (N.E.B.); McMullan-I2@ulster.ac.uk (I.L.M.); jj.wilson@ulster.ac.uk (J.J.W.); 2Sport and Exercise Sciences Research Institute, School of Sport, Newtownabbey BT37 0QB, UK; 3Institute of Mental Health Sciences, School of Health Sciences, Ulster University, Newtownabbey BT37 0QB, UK; 4The Cambridge Centre for Sport and Exercise Science, Anglia Ruskin University, Cambridge CB1 1PT, UK; Lee.Smith@aru.ac.uk; 5Institute of Public Health, City Exchange, 11–13 Gloucester St, Belfast BT1 4LS, UK; conor.cunningham@publichealth.ie; 6The Bamford Centre for Mental Health & Wellbeing Ulster University, Coleraine BT52 1SA, UK; Roger.OSullivan@publichealth.ie; 7Centre for Active and Healthy Ageing, Department of Sports Science and Clinical Biomechanics, University of Southern Denmark, Campusvej 55, 5230 Odense, Denmark; PCaserotti@health.sdu.dk

**Keywords:** chair-based exercise, physical function, older adults, systematic review

## Abstract

Physical activity is an important determinant of health in later life. The public health restrictions in response to COVID-19 have interrupted habitual physical activity behaviours in older adults. In response, numerous exercise programmes have been developed for older adults, many involving chair-based exercise. The aim of this systematic review was to synthesise the effects of chair-based exercise on the health of older adults. Ovid Medline, EMBASE, CINAHL, AMED, PyscInfo and SPORTDiscus databases were searched from inception to 1 April 2020. Chair-based exercise programmes in adults ≥50 years, lasting for at least 2 weeks and measuring the impact on physical function were included. Risk of bias of included studies were assessed using Cochrane risk of bias tool v2. Intervention content was described using TiDieR Criteria. Where sufficient studies (≥3 studies) reported data on an outcome, a random effects meta-analysis was performed. In total, 25 studies were included, with 19 studies in the meta-analyses. Seventeen studies had a low risk of bias and five had a high risk of bias. In this systematic review including 1388 participants, results demonstrated that chair-based exercise programmes improve upper extremity (handgrip strength: MD = 2.10; 95% CI = 0.76, 3.43 and 30 s arm curl test: MD = 2.82; 95% CI = 1.34, 4.31) and lower extremity function (30 s chair stand: MD 2.25; 95% CI = 0.64, 3.86). The findings suggest that chair-based exercises are effective and should be promoted as simple and easily implemented activities to maintain and develop strength for older adults.

## 1. Introduction

The importance of physical activity for older adults’ health is well documented [1,2]. Physical activity in later life reduces the risk of disease, helps to manage existing conditions and develops and maintains physical and mental function [1,2]. Inactive (defined as not meeting physical activity recommendations) older adults are at an increased risk of developing illnesses such as diabetes and cardiovascular disease [3,4,5], as well as experiencing a loss of mobility and functional independence due to declines in muscle mass, physical and neuromuscular function (e.g., muscle strength, power), and increased risk of sarcopenia [6].

This stage of life is an important period to promote physical activity to improve functions of daily living and slow progression of disease and disability. However, many older adults are not engaging in sufficient levels of physical activity to attain health benefits. In addition, the unprecedented government public health measures implemented to mitigate the spread of COVID-19 have had unintended negative consequences by further reducing older adults’ levels of physical activity [7] and increasing time spent in sedentary behaviours (sitting, reclining, lying down, watching television, reading, using mobile devices). These public health measures, which include social distancing and self-isolation for older adults, were implemented to shield older adults and vulnerable groups from the increased risk of more serious and potentially fatal illness associated with COVID-19 but caused abrupt and significant change to the networks and habits that many older adults established to maintain activity levels by restricting movement and recommending separation from work and family and friends. In addition, it is likely these restrictions have likely widened existing inequalities, whereby the physical activity of sub-groups of older adults, such as those who are socio-economically disadvantaged, frail, living with multi-morbidity or disability or living in residential care, may have been disproportionately affected.

The partial or total interruption to habitual physical activity can lead to deterioration in several metabolic and functional outcomes in older adults and there is concern that the regular and extended periods of lock-down may have increased risk for, and potentially worsened, existing chronic health conditions and caused acute and chronic deconditioning in the older adult population. Acute immobilization and bed rest studies represent extreme models of a lack of physical activity and provide a relevant insight into the plasticity of the neuromuscular system. Such studies indicate that in acute total absence of physical activity, neuromuscular function (e.g., muscle strength, explosive muscle force, muscle mass) is lost to a rate of up to over 3% per day starting nearly within the first few days [7,8,9]. However, to a lower extent, the consequences of COVID-19 may resemble such acute bed rest studies on deconditioning in older adults. In response, a number of public health and community and voluntary organizations have developed exercise resources for older adults, such as the Public Health England Active at Home campaign [10].

These programmes have typically focused on physical activities that can be easily implemented in the home with minimal equipment, such as chair-based exercise. Chair-based exercise is a seated, structured and progressive exercise programme, which uses a chair to provide stability and can be used by older adults and those who may be frail or deconditioned [11]. This type of exercise enables older adults to participate in safe, simple, and easily implemented physical activities [3]. For example, chair yoga enables individuals with declining mobility to participate in low-impact physical activity [12,13] and has been shown to be beneficial to psychological health in addition to improving mobility and physical function [12].

Given the increased time that many older adults are spending at home while complying with local and national regulations, the promotion of safe and effective activities that can be implemented in the home environment is both possible and necessary to maintain or increase their levels of physical activity. In light of this, there has been a call for research of the benefits of low-cost, high-reach initiatives to promote physical activity in those who are shielding from COVID-19 [14]. Current guidelines suggest that older adults should improve their physical function by undertaking activities aimed at improving or maintaining muscle strength, balance and flexibility activities on at least two days per week [15]. To our knowledge, only one systematic review of the effects of chair-based interventions has been published [3], which reviewed papers up to September 2017. This paper aims to update this previous review and extend the scope for a broader range of potential beneficiaries to systematically review the effects of chair-based exercise on the health of older adults.

## 2. Materials and Methods

This systematic review followed a pre-planned but unpublished protocol (available on request to corresponding author) and was conducted according to the PRISMA guidelines [16]. Ovid Medline, EMBASE, CINAHL, AMED, PyscInfo and SPORTDiscus databases were searched from inception to 1 April 2020. Search terms were based on a previous systematic review [3]. The search terms for Medline are included in Appendix A and were used to develop search strategies for the other databases. Following the removal of duplicates using Endnote (vX9.2, Thomson Reuters, PA, USA), titles and abstracts were independently screened for eligibility by two authors (NK, MAT) and any discrepancies were discussed by investigators for an agreed decision. A third author (NEB) was available to assist in resolving any discrepancies where required.

Studies were eligible if they included participants aged 50 years or older and living with or without a health condition. Eligible interventions consisted of primarily chair-based exercise for a minimum of two weeks. Only studies including a comparator group of non-seated exercise, usual care or no/minimal intervention were eligible for inclusion. Studies measuring physical function, using subjective or objective tools were included as they were identified as likely outcomes of chair-based exercise in a previous Delphi study [11].

The full text of the remaining articles was then assessed for inclusion independently by two authors (NK, MAT). Any discrepancies were discussed by investigators for an agreed decision. A third author (NEB) was available to assist resolving any discrepancies when required. Reference lists of relevant systematic reviews and included studies were hand searched for potentially eligible papers.

From the included studies, data were independently extracted by two authors (NK, MAT). Data extracted included the characteristics of studies including participants, study design, length of intervention, setting, country and adverse events. The interventions were described using the Template for Intervention Description and Replication (TiDieR) Criteria [17], including the following: why the intervention was being performed; what materials were used; who provided the intervention; how the intervention was delivered; how often and how long each session was. Lastly, we looked at modifications within the course of the intervention and how well the intervention was delivered according to the original intervention protocol and if adherence was assessed within each study.

### 2.1. Quality Assessment

Methodological quality of included studies was assessed using the Cochrane risk of bias tool v2 [18]. Studies were rated as having low, some concerns or high risk of bias for the randomization process, deviations from protocol, missing outcome data, measurement of the outcome and selective reporting. We considered that studies had a high risk of bias when at least one of the criteria was judged as having a high risk of bias. Overall risk of bias was assessed as having some concerns if one or more of the criteria were assessed as some concerns, but none were assessed as having a high risk of bias.

### 2.2. Data Synthesis

Where sufficient studies (≥3 studies) reported data on an outcome, a random effects meta-analysis was performed. Analyses were conducted on measures of balance, handgrip strength, timed up and go test, gait speed, 30-s arm curl test, 30-s chair stand, measures of activities of daily living and falls efficacy. As many studies did not report a change over time in outcomes, the difference between the intervention groups and control/comparison group at the latest available data point was included. Where there were more than one intervention group, data from the two groups were combined. The meta-analyses were conducted in RevMan Version 5.3 (The Cochrane Collaboration, Copenhagen, Denmark). Where outcomes were provided using the same instrument or were reported in comparable units, authors analyzed the mean difference (MD). Where outcomes were assessed using different measures, standardised mean difference (SMD) was assessed. Where data were not included in the meta-analyses, the effects were described narratively.

## 3. Results

### 3.1. Search Results

A PRISMA flowchart for the systematic literature search is included (Figure 1). Data bases and eligibility criteria were established and 20,539 articles were initially identified. Duplicate articles were removed, and the remaining 9694 articles were screened according to the inclusion criteria. An additional 9534 articles were excluded based on the title and abstract, and the full text of the remaining 160 articles were assessed for eligibility. Twenty-five studies were included in the review and were used in the meta-analysis, including 1388 participants. We found an additional five articles from the reference lists of included studies.

### 3.2. Characteristics of Included Studies

Six of the 25 included studies were based in Asia, five in the United States and four in Australia or New Zealand (Table 1). The remaining studies were in various European countries. Most (*n* = 19) studies were randomised controlled trials (RCTs). The studies included a wide variety of different populations from war veterans to patients with osteoarthritis or psychiatric disorders. There was also considerably heterogeneity in the age range of participants included. The mean age of participants in studies ranges from 55 [4] to 88 years [19].

Eleven studies included an alternative non-exercise control group, which included activities such as art or music therapy, health education or social activities. Ten studies included non-intervention control groups, where participants were asked to maintain everyday activity or continue with usual treatments. Other studies offered alternative exercise, such as information on nearby yoga classes, resistance exercise, limb mobilization and balance training in a standing position (Table 1).

The most common objective measures of physical function were measures of balance (Berg Balance Scale (*n* = 7/25 studies)), upper limb function and strength (handgrip strength (*n* = 7/25 studies), and 30 s arm curl test (*n* = 3/25 studies)) or mobility (timed up and go test (*n* = 7/25 studies) or gait speed (*n* = 6/25 studies)), activities of daily living (*n* = 4/25 studies) or falls efficacy (*n* = 5/25 studies), and these were included in the meta-analysis. There were a wide range of other tests (listed in Table 1), which included various tests of balance, mobility and upper or lower limb strength and flexibility. Similarly, there were a wide range of subjective measures of physical function. Common among these were the falls efficacy scale or assessments of activities of daily living (Table 1).

### 3.3. Intervention Characteristics

Interventions sought to promote either aerobic (*n* = 3), strength (*n* = 6) or flexibility and range of motion (*n* = 7) (Table 2). It should be noted that not all studies included components targeting upper limb function or balance. Seven studies included chair-based yoga, one used seated tai chi and one used a rocking chair. Along with using a chair to perform the exercise, weights, balls and music were also used to enrich the participants’ experience, as well as improving range of motion. Most of the classes were carried out by an instructor in a group setting. Several studies also used photos, booklets and DVDs to enable participants to perform exercises in their own home. The interventions lasted between two [21] and 72 weeks [32], with the most common duration being 12 weeks, delivering two to 14 sessions per week (mode = 2). Sessions lasted between 15 [33] and 110 [12] minutes (mode = 45 min) (Table 2). Eleven of the 25 studies assessed adherence, with three using participant logs or diaries [32,35,36,42]. However, only nine of these reported actual adherence, ranging from 70% [41] to 96% [33] completion of prescribed sessions.

### 3.4. Risk of Bias

Using a Cochrane Risk of Bias tool v2 [18], the authors assessed the risk of bias of the included studies (Figure 2). The overall risk of bias was low for 17 studies, and three were assessed as unsure [19,23,31] because of a lack of clarity in the randomization process. Five studies were assessed as having a high overall risk of bias [12,22,24,32,39]. Studies with a high risk of bias were classified as such based on their lack of randomization in the design. Details of risk of bias for each study are included in Appendix A.

### 3.5. Effects on Physical Function

Nineteen studies provided data for use in meta-analyses (Table 3, Appendix A). Most studies reported the differences at follow-up and no change over time. Chair-based exercise led to improvements in handgrip strength (MD = 2.10; 95% CI = 0.76, 3.43; I2 = 42%) in seven studies including a total of 266 participants. Significant improvements were also observed for 30 s arm curl test (MD = 2.82; 95% CI = 1.34, 4.31; I2 = 71%) and 30 s chair stand test (MD 2.25; 95% CI = 0.64, 3.86; I2 = 62%), but the high heterogeneity in these outcomes should be noted. No significant differences between groups were observed in the Berg balance scale, timed up and go test or gait speed between the intervention and control groups. Similarly, no significant differences were observed for self-reported activities of daily living or for falls efficacy, which were analyzed using standardised mean difference between the intervention and control groups as there were different instruments used to measure each outcome.

There was insufficient data to include other physical function outcomes in these meta-analyses. In keeping with the significant findings in the meta-analyses, upper limb strength and flexibility were seen to improve in the intervention group in the studies by Yao et al. [12] and Venturelli et al. [40]. In addition, some studies showed improvements in lower limb strength and flexibility. Lower limb muscle endurance improved at the end of the intervention and six weeks after the intervention had ended in Ikai et al.’s [4] study. Niemelä et al. [33] and McMurdo et al. [30] demonstrated significant improvements in the intervention group compared to control in knee extension strength and quadriceps strength, respectively. Rieping et al. [39] and Yao et al. [12] demonstrated that lower limb muscle strength improved in the intervention group. In a study of participants who had recently suffered a stroke, Dean et al. [21] showed an improvement in peak vertical force through the affected foot during standing in the intervention, up by 21% of body weight (95% CI 14 to 28) compared with the control group.

Benefits of the intervention were also demonstrated on dynamic measures of balance [12]. A number of studies used multi-component tests that incorporated various domains of physical function. Flexibility was measured as the maximum seated reach distance [21,28] in a sit and reach test [20], or using a body anti-flexion measuring device [4] or spinal flexion [29]. Baum et al. [19] demonstrated that chair-based exercise for 26 weeks led to significant increases in the physical performance test. Similar findings were demonstrated by Vogler et al. [41] from the physical performance and mobility examination, by Daniel et al. [20] from the senior fitness test and by Park et al. [37] from the manual functional test.

## 4. Discussion

In this systematic review including 1388 participants, results demonstrated that chair-based exercise programmes improve upper extremity (handgrip strength and 30 s arm curl test) and lower extremity (30 s chair stand) function. These changes were observed in short (<12 weeks) and medium term (12 weeks to 6 months) interventions. Only one study examined the longer term impact of chair-based exercise, showing no differences in grip strength or upper limb range of motion after 18 months in the intervention group (*n* = 20) compared to the control group (*n* = 10) of older adults following a hip fracture [32].

The age related decline in upper extremity function, such as handgrip strength, affects everyday function, such as the ability to hold heavy objects. Our recent umbrella review with integrated meta-analyses of the health outcomes associated with handgrip strength demonstrated that having a higher grip strength was associated with a reduced risk of early mortality, cardiovascular disease and disability [43]. It is also a good indicator of biological ageing, whereby the bodies systems are ageing faster than average for a person of a similar age [44]. Therefore, the statistically significant impact of chair-based exercise is an important finding. As the difference between the groups at the end of the intervention was within the range of what would be a minimal clinically important difference in clinical populations [45], it demonstrates the potential clinical significance of the findings too.

Differences in both the 30-s chair stand and arm curl tests demonstrated improvements in neuromuscular function (e.g., strength) as a result of chair-based exercise. The difference observed in the 30-s chair stand was of a similar magnitude to that demonstrated by [46] when they compared high and low active community-dwelling older adults. The findings of a lack of effect on balance are in keeping with a previous review of chair-based exercise [3]. Changes in aerobic physical activity in older adults have been shown to improve balance [47].

The current paper adds to the previous review of seated exercise [3] by updating the search to include the last three years of evidence and broadening the inclusion criteria to include all groups of older adults, not just those living with a health condition or impairment. This reflects the current situation whereby chair-based exercise is being recommended for all older adults whilst COVID-19 public health measures have been in place.

The public health restrictions in place to prevent the spread of transmission have impacted on physical activity levels, and there have been calls to focus on supporting older adults to meet the recommended levels of physical activity [48]. However, given the closure of leisure centres and recreation facilities, many programmes recommended chair-based exercises. The evidence from our review indicates that the benefits from these programmes may be limited in scope. Future programmes should follow the physical activity recommendations that older adults should aim to engage in at least two sessions of strength, balance and flexibility exercise per week, in addition to at least 150 min of moderate intensity activity per week [15].

This review was completed according to PRISMA guidelines. A systematic search strategy was used; all studies were independently screened for inclusion and data extraction was completed by two independent researchers. Given the expected heterogeneity, a conservative random-effects model was applied to all meta-analyses. However, the heterogeneity in the intervention components and included populations should be noted. For example, Nicholson et al. [32] noted the effects of the intervention may be obscured by the heterogeneity of participants. This is a reminder that not all older adults are similar in terms of functional ability and health status, and programmes should be tailored to meet these needs. Five of the 25 studies were assessed as having a high risk of bias as they did not employ randomization to allocate participants to the intervention or control groups. Only 11 of the 25 studies reported recording adherence, and only nine of these reported the actual adherence. Future studies should include a process evaluation to explore the fidelity of the intervention. A final limitation to note is that all of the included studies were published in English. We did not exclude any studies based on language and attempted to translate titles and abstracts to check for eligibility to mitigate against this risk. The recommendations by Sexton et al. [3] to improve the methodological quality of future research, such as increased sample size and the quality of the interventions, such as improving progression plans for interventions, remain as gaps in this updated literature.

## 5. Conclusions

This review highlights that chair-based exercise benefits several aspects of physical function in older adults. Balance, gait speed, grip strength and several other physical measurements were often documented as improved in individuals who engaged in chair-based exercise. These findings add to a growing body of evidence that supports the importance of both light intensity activity for health and strength and balance activities to preserve physical function, a message that is particularly important for those who are currently inactive, and as such, chair-based exercise can be promoted as a safe and progressive mode of activity for those who may be frail or deconditioned.

In addition, the evidence in this review was mainly of good quality (low risk of overall bias), suggesting that chair-based exercises should be promoted as simple and easily implemented activities to maintain and develop strength and offset the negative effects of physical inactivity in older adults and vulnerable populations who may be self-isolating during the pandemic. In this respect, dissemination of easily understandable information (by governments, public health agencies, health professionals and community-based organizations) is critical to ensuring that older people have clear messages and resources on how to integrate chair-based activity into the home environment to stay physically and mentally healthy at this time.

In communicating the benefits of chair-based activities public health messaging should reinforce the evidence that every minute counts: any activity is better than none, and everyone (all ages and abilities) should aim to move more and move more often [15], whilst also adhering to the important, but often neglected, guidance to engage in strength and balance exercise. This messaging will be particularly important going forward, and it is imperative that policy and practice support all members of society to achieve the recommended levels of physical activity to ensure that they are not disadvantaged in the short or long term by COVID-19.

## Figures and Tables

**Figure 1 ijerph-18-01902-f001:**
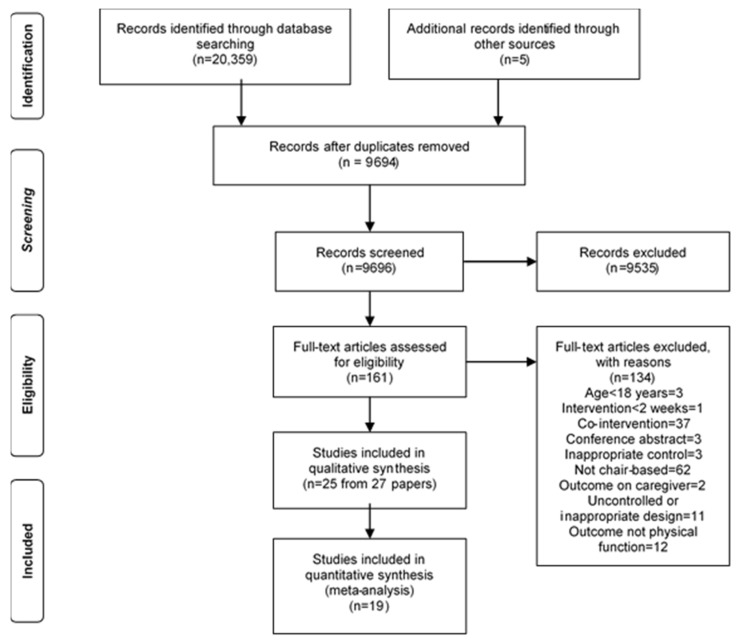
PRISMA 2009 Flow Diagram.

**Figure 2 ijerph-18-01902-f002:**
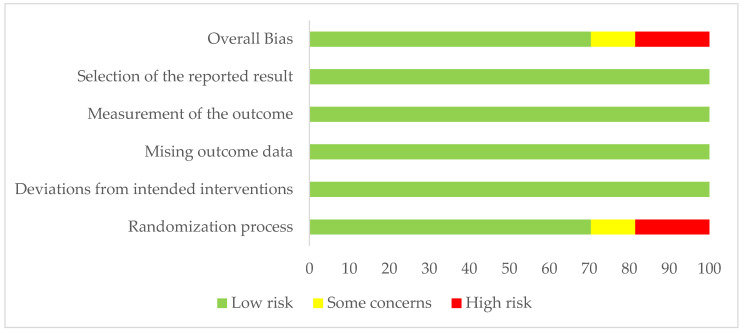
Risk of bias of included studies.

**Table 1 ijerph-18-01902-t001:** Characteristics of Included Studies.

Author (Year)	Sample Size	Country	Age Mean (SD)	Gender	Study Design	Control Group	Participants’ Characteristics	Physical Function Outcomes
Baum (2003) [19]	20	USA	88 (NR)	NR	RCT	Art therapy	Frail elderly	Timed up and go, physical performance test, Berg balance
Daniel (2012) [20]	15	USA	77 (5.3)	M = 39%, F = 61%	RCT	Non-intervention control	Pre-frail older adults	Chair stands, timed up and go, timed arm curls, sit and reach, step 2 test, 6-min walk test, senior fitness test, fear of falling, LLFDI-function
Dean (2017) [21]	12	Australia	67 (11.9)	M = 64%, F = 36%	RCT	Seated cognitive-manipulative tasks	Stroke survivors	Sitting ability, sitting quality, 10-m walk test
Furtado (2016) [22]	35	Portugal	83.81 (6.6)	M = 0%, F = 100%	CBA	Non-intervention control	Institutionalised older adults	Lawton Instrumental Activities of Daily Living, Tinetti Falls Efficacy Scale, senior fitness test, 30 s chair sit to stand, 30 s arm-curl test, chair sit-and-reach test, timed up and go
Furtado (2020) [23]	40	Portugal	81.62 (7.91)	M = 0%, F = 100%	RCT	Non-intervention control	Institutionalised older women	Physical frailty, Falls Efficacy Scale, Katz Index of Independence in ADL
Ikai (2017) [4]	56	Japan	55.3 (13.7)	M = 64%, F = 36%	RCT	Instructed to spend group time in activities of their choice (e.g., walking, reading or chatting)	Inpatients with chronic psychiatric disorders	Anteflexion in sitting, handgrip, modified falls efficacy scale, postural sway
Kertapati (2018) [24]	84	Indonesia	66 (NR)	M = 19%, F = 81%	CBA	Non-intervention control	Older adults living in in Depok City, Indonesia	Functional Independence Measure
Kim (2015) [25]	30	Korea	73.2 (3.1)	M = 0%, F = 100%	CBA	Standing balance training	Community-dwelling women aged ≥65 years	Short-form berg balance, 10-m walk test, timed up and go, falls efficacy scale
Kujasski (2018) [26]	55	Poland	65 (NR)	M = 9%, F = 91%	RCT	Resistance exercise programme	Community-dwelling older adults aged ≥55 years	6-min walk test
Latham (2003) [27]	242	New Zealand and Australia	79.1 (6.9)	M = 47%, F = 53%	RCT	Offered general advice on problems encountered	Frail older adults after hospital discharge	ADL (Barthel index), modified falls efficacy scale, quadriceps strength, time taken to walk 4 m, timed up and go, berg balance
Lee (2015) [28]	59	Hong Kong	85.8 (9.2)	M = 23.7%, F = 76.3%	RCT	Limb mobilization programme	Living in residential care facilities	Sequential weight shifting, forward reach, eye–hand coordination
McMurdo (1993) [29]	41	Scotland	80.4 (6.5)	M = 89%, F = 11%	RCT	Reminiscence sessions without exercise	Older adults living in residential homes	Sway, handgrip strength, chair-to-stand, ADL (Barthel index)
McMurdo (1994) [30]	65	Scotland	82.9 (6)	M = 83%, F = 17%	RCT	Reminiscence sessions without exercise	Older adults living in residential homes	Quadriceps strength, step test
Netz (2007) [31]	26	Israel	76.9 (6.72)	M = 48%, F = 52%	RCT	Social activity including conversations, social games, viewing pictures and reading newspaper articles	Older adults with dementia	Timed up and go, sit to stand, functional reach
Nicholson (1997) [32]	30	South Africa	79.8 (6.6)	M = 0%, F = 100%	CBA	Non-intervention control	Patients discharged from an orthopaedic ward 8–10 days after hip surgery	Handgrip strength, Falls Efficacy Scale
Niemela (2011) [33]	51	Finland	80.2 (3.6)	M = 0%, F = 100%	RCT	Non-intervention control	War veterans, spouse, or widow of war veterans	Max walking speed, handgrip, chair rising, standing on one leg, Berg balance scale
Park (2014) [34]	25	USA	79 (6.42)	M = 23.5%, F = 76.5%	RCT	Health Education programme	Osteoarthritis	Gait speed, 6-min walk test, Berg balance scale
Park (2016 and 2017a) [35,36]	112	USA	75.3 (7.5)	M = 24.1%, F = 75.9%	RCT	Health Education programme	Lower Extremity Osteoarthritis	Gait speed, Berg balance scale
Park (2017b) [37]	26	Korea	NR	NR	RCT	Conventional physiotherapy	Stroke patients	Manual Function Test, handgrip strength, berg balance scale, postural sway, time taken to walk 10 m
Park (2019) [38]	31	USA	84.3 (7.7)	M = 58.1%, F = 41.9%	Cluster RCT	Music therapy groups	Individuals living with dementia	Timed up and go, physical performance test, SPPB, handgrip
Rieping (2019) [39]	32	Portugal	80 (8.04)	M = 0%, F = 100%	CBA	Non-intervention control	Institutionalised Older Women	30-s arm-curl test, 30-s chair seat and stand test, 8ft up and go test, Falls Efficacy Scale, Lawton Scale of Instrumental ADL, Katz Index of Independence in ADL
Venturelli (2010) [40]	30	Italy	84 (6)	M = 0%, F = 100%	RCT	Usual care	Frail women	Arm curl strength test, ADL (Barthel index), performance-oriented mobility assessment index
Vogler (2009) [41]	120	Australia	80 (7)	M = 21%, F = 79%	RCT	Social visit by a research assistant at the same frequency as the exercise group	Inpatients in a care and rehabilitation facility	Physical Performance and Mobility Examination, maximal balance range tests
Vogler (2012) [42]	120	Australia	80 (7)	M = 21%, F = 79%	RCT	Social visit by a research assistant at the same frequency as the exercise group	Older people recently discharged from hospital	Physiological Profile Assessment, maximal balance range tests
Yao (2019) [12]	31	Taiwan	77.5 (6.2)	M = 0%, F = 100%	CBA	Maintained regular daily activities	Community dwelling older females	Handgrip strength, lower limb muscle strength, upper limb muscle strength, static balance, agility, dynamic balance, lower limb flexibility, upper limb flexibility

SD = standard deviation, USA = United States of America, RCT = randomised controlled trial, m = male, f = female, LLFDI = late life function and disability instrument, m = metre, ADL = activities of daily living, CBA = controlled before and after, s = second, NR = not reported, ADL = activities of daily living, mon = minute, SPPB = short physical performance battery.

**Table 2 ijerph-18-01902-t002:** Characteristics of included interventions according to Template for Intervention Description and Replication (TiDier) criteria.

Author (Year)	Brief Description of Intervention	Who Delivered the Intervention	Mode of Delivery	Where Exercise Took Place	Length of Intervention (Weeks)	Total Number of Sessions	Frequency of Sessions (Per Week)	Duration of Sessions (mins)	How Adherence Was Assessed	Adherence
Baum (2003) [19]	Chair-based exercise with weights	Exercise physiologist	Group exercise	Long term care facility	26	78	3	60	NR	NR
Daniel (2012) [20]	Seated aerobic exercises	Certified fitness professional	Group exercise	Study site	15	45	3	45	Attendance at sessions	86%
Dean (2017) [21]	Seated reaching tasks	NR	NR	Hospital rehabilitation facility	2	10	5	30	NR	NR
Furtado (2016) [22]	Chair yoga based on hatha yoga, focusing on flexibility	Expert technicians	NR	Social and health care support centres	14	28	2	NR	Attendance at sessions	NR
Furtado (2020) [23]	Chair exercises with TheraBand	Instructor	Group exercise	Gym	28	84	3	45	Attendance at sessions	72%
Ikai (2017) [4]	Chair yoga	Yoga instructor and occupational health staff support	Group exercise	Hospital	12	24	2	20	NR	NR
Kertapati (2018) [24]	Chair yoga with spiritual intervention	Instructor	Group exercise	NR	4	12	3	60	NR	NR
Kim (2015) [25]	Seated stretching	NR	NR	NR	8	24	3	20	NR	NR
Kujasski (2018) [26]	Seated stretching and mobility exercises	NR	NR	University campus	12	24	2	45–50	Technician recorded attendance	71%
Latham (2003) [27]	Quadriceps exercises using adjustable ankle cuff weights	Physical therapists	Group and individual exercise	Hospitals/home	10	30	3	NR	NR	NR
Lee (2015) [28]	Seated tai chi	Researchers who developed the exercise	Group exercise	Residential care facilities	12	36	3	60	NR	NR
McMurdo (1993) [29]	Upper and lower limb flexibility and strengthening exercises while seated	NR	Group exercise	Local authority residential homes	28	56	2	45	Attendance at sessions	91%
McMurdo (1994) [30]	Isometric exercises to music designed to strengthen major muscle groups and improve joint flexibility and muscle tone	Physiotherapist	Group exercise	Local authority residential homes	24	48	2	45	Attendance at sessions	72%
Netz (2007) [31]	Seated exercises to promote range of motion, strength and coordination of upper and lower limbs	Physical activity leader	Instructor-patient interaction	Day centre	12	24	2	45	NR	NR
Nicholson (1997) [32]	Seated exercise with simple objects to help patients’ full range of motion	Physiotherapist	Group exercise	Geriatric hospital	72	24	NR	50	NR	NR
Niemela (2011) [33]	Rocking chair exercises	Physiotherapist	Individual exercises at home	Rehab centre and home	6	84	14	15	Adherence rates obtained from diaries	96%
Park (2014) [34]	The Sit ‘N’ Fit Chair Yoga programme	Yoga instructor	Instructor	Senior centre	8	16	2	45	NR	NR
Park (2016 and 2017a) [35,36]	Chair yoga	Certified yoga instructor	Group exercise and instruction manual with photos	Senior housing development	8	16	2	45	Daily logs	Logs were not completed
Park (2017b) [37]	Sitting boxing	NR	Group exercise	Rehabilitation care hospital	6	18	3	30	NR	NR
Park (2019) [38]	Chair yoga	Certified yoga instructor, music therapist, fitness instructor	Group exercise	College of medicine and centre for comprehensive brain health, community-based day centres for AD/dementia	12	24	2	45	NR	NR
Rieping (2019) [39]	Chair-based exercises: aerobic or with TheraBand	NR	NR	NR	14	28	2	45	NR	NR
Venturelli (2010) [40]	Circuit-based upper body exercise	Kinesiologist	Group exercise	Residents from a geriatric institute	12	36	3	45	Attendance at sessions	75%
Vogler (2009) [41]	Seated exercises targeted hip flexion, extension, abduction, knee flexion and extension, and ankle plantar and dorsiflexion	Physical therapists	Delivered at participants’ home	Aged care and rehabilitation centre- Hospital/home	12	36	3	NR	Attendance at sessions	70%
Vogler (2012) [42]	Home-based seated exercises: weight bearing or resistance	Physiotherapists	Group exercise	Home	12	36	3	NR	Self-reported completion of exercise sessions	70%
Yao (2019) [12]	Chair yoga	Instructor	Group exercise	2 communities	12	24	2	110	NR	NR

NR = Not reported.

**Table 3 ijerph-18-01902-t003:** Results from meta-analysis.

Outcome	Effect Size	95% CI	No. of Participants	No. of Studies	I^2^ (%)
**Objective Physical Function**					
**Berg Balance Scale**	MD 0.66	−1.01, 2.33	359	5	20%
**Handgrip**	MD 2.10 *	0.76, 3.43	266	7	42%
**Timed up and go test**	MD 0.95	−1.12, 3.01	394	7	39%
**Gait Speed**	MD −0.03	−0.21, 0.16	450	6	77%
**30 s arm curl test**	MD 2.82 *	1.34, 4.31	97	3	71%
**30 s chair stand test**	MD 2.25 *	0.64, 3.86	97	3	62%
**Subjective Physical Function**					
**Activities of daily living**	SMD 0.32	−0.4, 1.04	126	4	74%
**Falls Efficacy**	SMD −0.06	−0.46, 0.34	208	5	49%

MD = mean difference, SMD = standardised mean difference. * *p* < 0.05.

## Data Availability

Not applicable.

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
