# Peer review of "The Effect of Chair-Based Exercise on Physical Function in Older Adults: A Systematic Review and Meta-Analysis"

_ijerph, 2021, doi:10.3390/ijerph18041902_

Round 1

Reviewer 1 Report

The authors present a systematic literature review following the PRISMA guidelines. The title and introduction are associated with the topic at hand, which is of current, high relevance. Most included studies in the SLR were RCT's which add a great value.

I felt the methods section did not quite reach repeatability. It was not clear until reading the supplementary materials how the subgroups were built and compared. Therefore, when I reached section 3.5, I was pleasantly surprised that the authors disclosed that some test comparisons had high heterogeneity. However, I felt a bit misled since no previous explanation was given. 

I suggest clarifying the high variability in the source populations and the age since mean ages varied from the early 40's to the late '80s. I was a bit lost when reading and trying to understand the first paragraph on page 5/18. Unsure if all the groups added up or went over the count of studies included. Furthermore, expand the explanation that not all studies included interventions to increase handgrip strength or postural balance and so forth. That also misled me. 

The study is meritorious of publication after minor corrections.

Author Response

Response to Reviewer 1

Comment 1: The authors present a systematic literature review following the PRISMA guidelines. The title and introduction are associated with the topic at hand, which is of current, high relevance. Most included studies in the SLR were RCT's which add a great value.

Response 1: Thank you for this positive response.

Comment 2: I felt the methods section did not quite reach repeatability. It was not clear until reading the supplementary materials how the subgroups were built and compared. Therefore, when I reached section 3.5, I was pleasantly surprised that the authors disclosed that some test comparisons had high heterogeneity. However, I felt a bit misled since no previous explanation was given. 

Response 2: We appreciate this interesting comment. We grouped the analyses by outcome. Within our meta-analyses (& forest plots in supplementary analyses), we do not present subgroup analyses. We have clarified in section 2.2 that the synthesis was conducted on outcomes grouped by different measures of physical function.

Comment 3: I suggest clarifying the high variability in the source populations and the age since mean ages varied from the early 40's to the late '80s. I was a bit lost when reading and trying to understand the first paragraph on page 5/18. Unsure if all the groups added up or went over the count of studies included. Furthermore, expand the explanation that not all studies included interventions to increase handgrip strength or postural balance and so forth. That also misled me. 

Response 3: Thank you for this comment. We have noted the heterogeneity in the age range of participants in section 3.2. We have also added to the previous description of the outcomes to note the number of studies that each test was used in. In section 3.3, we have also noted that not all interventions aimed to increase handgrip strength or balance. We trust this provides the clarity required.

Comment 4: The study is meritorious of publication after minor corrections.

Response 4: Thank you

Reviewer 2 Report

To authors

This study is considered to be an excellent a systematic review and meta-analysis of exercise effects on physical function in the elderly.

I think that this study will be better if the following two points are supplemented.

1.

Enter the systematic review registration number in “Abstract”.

2.

Please include more detailed Limitations in “Discussion”: Discuss limitations at study and outcome level (e.g., risk of bias), and at review-level (e.g., incomplete retrieval of identified research, reporting bias.

Hopefully my advice will improve the quality of your research.

Author Response

Response to Reviewer 2

Comment1: To authors

This study is considered to be an excellent a systematic review and meta-analysis of exercise effects on physical function in the elderly. 

I think that this study will be better if the following two points are supplemented.

Response 1: Thank you for your review and positive comments

Comment 2: 1. Enter the systematic review registration number in “Abstract”.

Response 2: As noted in section 2 (Materials and Methods), the systematic review followed a pre-planned but unpublished protocol. Therefore, a registration number is not available.

Comment 3: 2. Please include more detailed Limitations in “Discussion”: Discuss limitations at study and outcome level (e.g., risk of bias), and at review-level (e.g., incomplete retrieval of identified research, reporting bias.

Response 3: We have noted more detailed limitations at the end of the discussion to highlight that the lack of randomisation is a limit at the outcome level and retrieving only English language studies as a limit at the review level.

Comment 4: Hopefully my advice will improve the quality of your research.

Response 4: Thank you for the helpful points.

Reviewer 3 Report

This is a well written and presented article on a topic of public health importance. The proliferation ‘home-based’ activity that has resulted from Covid-19 restrictions warrants an exploration of the extant evidence-base, particularly amongst vulnerable populations. The authors should be congratulated for proactively addressing this gap. Furthermore, the importance of strength and balance activities for older adults is a public health communication priority and the evidence synthesised here, underscores this.

The introduction offers a strong rationale for the study and I support the urgency to address the ‘inactivity conditions’ created by lockdown, however, these conditions have not affected people equally – including older adults. There are significant inequalities within age-defined populations and I would encourage the authors to consider this within their opening statements. Specifically, I’d like to see some consideration given to factors such as disadvantage, frailty, multi-morbidity (including mental health conditions), disability and older adults living in supported housing or care homes. How has lockdown impacted these populations activity and what might this mean for who we might wish to target with chair-based intervention. This is important, as the greatest effect of physical activity can be had in the least active and high risk populations.

The methods are concise and I have few concerns regarding rigour, however, I would like the authors to provide a rationale for the inclusion of study 39 and study 41 – wheelchair bound with spinal cord injuries - with a sample mean age below the inclusion criteria. These studies stand out to me as not appropriate given the context of the review and this does raise a red flag about the screening process. I was encouraged to see use of the TiDieR framework to describe study components.

The reporting of the results would benefit from further comment on the adherence to protocol and the reporting of adherence across studies – which is poor. This is an important point to draw out and I would also like to see this explored fully in the discussion and as a filter for the outcomes reported.

I recognise that the focus of the paper is on physical function but could the authors perhaps explain why they did not explore the impact of chair-based exercise on quality of life or psychological wellbeing outcomes. These outcomes seem particularly relevant given that the authors set the context of the review within lockdown restrictions. Is a separate review planned that addresses these outcomes perhaps?

Authors note that …not all older adults are similar in terms of functional ability and health status and programmes should be tailored to meet these needs.What does this review tell us or rather what can be argued about the role of Chair-based exercise for these different within population characteristics? I think this is an important element of intervention specificity and ultimately effect and links to my points about the introduction to the paper.

I’d also like to see fuller discussion on what this review adds to previous reviews (i.e. Sexton et al., 2018). Consideration is limited to a comment on the lack of effect on balance. How, if at all, has the evidence for chair-based exercise advanced? What is next and where can improvements in method, measurement of outcomes, study design and intervention delivery and reporting be made? The review offers little in terms of areas for future research for example.

The concluding statements are positive about the value of chair-based exercise (e.g. the evidence in this review suggests that chair-based exercises should be promoted) and yet this seems a little at odds with the following statement in the discussion ‘The evidence from our review indicates that the benefits from these programmes may be limited’. Whilst I welcome the statements from the authors about the importance of clarity of communication and messaging, could I suggest that the clarity of the message regarding the strength of the evidence for chair-based exercise is revisited and re-framed. Linked to this, I would like the authors to explore whether chair-based exercise should be primarily promoted to address the need for strength and balance activities in older adults OR whether a focus on ‘light-intensity aerobic’ activity is where the messaging is best placed. It might of course offer an opportunity to do both depending on the dose and design of intervention. This is worthy of brief comment as the current conclusions confuse these messages somewhat.

Author Response

Response to Reviewer 3

Comment 1: This is a well written and presented article on a topic of public health importance. The proliferation ‘home-based’ activity that has resulted from Covid-19 restrictions warrants an exploration of the extant evidence-base, particularly amongst vulnerable populations. The authors should be congratulated for proactively addressing this gap. Furthermore, the importance of strength and balance activities for older adults is a public health communication priority and the evidence synthesised here, underscores this.

Response 1:  Thank you. We agree this is an important gap that needs addressed and would like to sincerely thank Reviewer 3 for the helpful minor amendments they have suggested. We believe this have improved the article further.

Comment 2: The introduction offers a strong rationale for the study and I support the urgency to address the ‘inactivity conditions’ created by lockdown, however, these conditions have not affected people equally – including older adults. There are significant inequalities within age-defined populations and I would encourage the authors to consider this within their opening statements. Specifically, I’d like to see some consideration given to factors such as disadvantage, frailty, multi-morbidity (including mental health conditions), disability and older adults living in supported housing or care homes. How has lockdown impacted these populations activity and what might this mean for who we might wish to target with chair-based intervention. This is important, as the greatest effect of physical activity can be had in the least active and high risk populations.

Response 2: Thank you for your favourable comments. We agree that lockdowns are likely to have widened inequalities in sub-groups of older adults, including this identified. It appears logical that the effect would be greatest in those who are socio-economically disadvantaged, frail and those living with multi-morbidity or disability. However, there is a limited of evidence in the peer-reviewed literature for these groups. In the last two weeks we have had a systematic review of the impact of the pandemic on physical activity published (BMJ Open Sport & Exercise Medicine 2021;7:e000960). We did not identify published studies of the effects of the pandemic on physical activity in these sub-populations. Therefore, in response to this comment we have highlighted in the introduction the possibility of an inequitable effect in specific populations.

Comment 3: The methods are concise and I have few concerns regarding rigour, however, I would like the authors to provide a rationale for the inclusion of study 39 and study 41 – wheelchair bound with spinal cord injuries - with a sample mean age below the inclusion criteria. These studies stand out to me as not appropriate given the context of the review and this does raise a red flag about the screening process. I was encouraged to see use of the TiDieR framework to describe study components.

Response 3: We appreciate the careful and considered review that this author has provided. Our selection criteria were pre-specified and did not exclude studies of individuals with any existing medical condition. Therefore, we could not exclude these two studies based on this criteria. We do regret the oversight regarding age, and have now excluded them from the review based in this. We have changed the text throughout to reflect this. The overall findings have not been affected by this.

Comment 4: The reporting of the results would benefit from further comment on the adherence to protocol and the reporting of adherence across studies – which is poor. This is an important point to draw out and I would also like to see this explored fully in the discussion and as a filter for the outcomes reported.

Response 4:  We agree that adherence is poorly reported, and have noted the reporting of adherence in the results and as a limitation in the discussion. Given the poor reporting, it would not be a reliable filter for the results at we cannot assume from unreported information that adherence was poor.

Comment 5: I recognise that the focus of the paper is on physical function but could the authors perhaps explain why they did not explore the impact of chair-based exercise on quality of life or psychological wellbeing outcomes. These outcomes seem particularly relevant given that the authors set the context of the review within lockdown restrictions. Is a separate review planned that addresses these outcomes perhaps?

Response 5: We agree that psychological outcomes and quality of life are important outcomes. We are planning a separate review on this topic.

Comment 6: Authors note that “…not all older adults are similar in terms of functional ability and health status and programmes should be tailored to meet these needs.” What does this review tell us or rather what can be argued about the role of Chair-based exercise for these different within population characteristics? I think this is an important element of intervention specificity and ultimately effect and links to my points about the introduction to the paper.

Response 6:  We did not identify any specific information in this review that would provide the answer to this question, but felt it was important to at identify the principle that different groups of older adults have differing needs from a programme.

Comment 7: I’d also like to see fuller discussion on what this review adds to previous reviews (i.e. Sexton et al., 2018). Consideration is limited to a comment on the lack of effect on balance. How, if at all, has the evidence for chair-based exercise advanced? What is next and where can improvements in method, measurement of outcomes, study design and intervention delivery and reporting be made? The review offers little in terms of areas for future research for example.

Response 7:  We have added to the discussion of how the current review adds to those that have been conducted previously. One important addition is the inclusion of a broader range of participants. The previous Sexton review included only older adults living with a health condition or impairment. However, given that chair-based exercise is being recommended to all older adults, we felt it was important to include a wider range of participants. In addition, we have updated the search which was conducted three years before the current review.

Comment 8: The concluding statements are positive about the value of chair-based exercise (e.g. the evidence in this review suggests that chair-based exercises should be promoted) and yet this seems a little at odds with the following statement in the discussion ‘The evidence from our review indicates that the benefits from these programmes may be limited’. Whilst I welcome the statements from the authors about the importance of clarity of communication and messaging, could I suggest that the clarity of the message regarding the strength of the evidence for chair-based exercise is revisited and re-framed.

Response 8: We apologies for the lack of clarity. What we were trying to communicate was that the ‘range’ of benefits may be limited in the second quote. We have added clarity to this statement in the text. We have also noted the strength of evidence in the conclusion to support the assertion that chair-based exercise should be promoted.

Comment 9: Linked to this, I would like the authors to explore whether chair-based exercise should be primarily promoted to address the need for strength and balance activities in older adults OR whether a focus on ‘light-intensity aerobic’ activity is where the messaging is best placed. It might of course offer an opportunity to do both depending on the dose and design of intervention. This is worthy of brief comment as the current conclusions confuse these messages somewhat.

Response 9: We agree with this assessment that chair-based exercise could offer an opportunity for both light intensity activity and strength and balance for older adults. We have tried to preserve the succinctness of the messages in the conclusion, and have therefore added a note that both light intensity activity and strength and balance activities are important.